# High Martensitic Steel after Welding with Micro-Jet Cooling in Microstructural and Mechanical Investigations

**DOI:** 10.3390/ma14040936

**Published:** 2021-02-16

**Authors:** Bożena Szczucka-Lasota, Tomasz Węgrzyn, Tadeusz Szymczak, Adam Jurek

**Affiliations:** 1Faculty of Transport and Aviation Engineering, Silesian University of Technology, Krasińskiego 8, 40-119 Katowice, Poland; bozena.szczucka-lasota@polsl.pl; 2Department of Vehicle Type-Approval & Testing, Motor Transport Institute, ITS, Jagiellońska 80, 03-301 Warsaw, Poland; tadeusz.szymczak@its.waw.pl; 3Novar Sp. z o. o., Towarowa 2, 44-100 Gliwice, Poland; adam.jurek@novar.pl

**Keywords:** smart city, transport, vehicles, mechanical engineering, Docol steel, micro-jet welding, mechanical tests, microstructure, mini-specimen, fracture, fatigue

## Abstract

Modern means of transport will play a significant role in the smart city. In the automotive industry, high-strength steels such as Docol are employed more often. This kind of material is relatively not very well weldable. The main reason is related to the Heat Affect Zone, the region in which cracks occur. Another disadvantage is connected with differences in values of ultimate strength of parent and weld material. The differences can be diminished using the correct welding process, which employs nickel and molybdenum electrode wires at much lower sulfur content. The weld metal deposit contains mainly martensite and bainite with coarse ferrite, while the parent material contains mainly martensite and rather fine ferrite. New technology, micro-jet cooling after the joining process enables to obtain the microstructure of weld metal deposit at acceptable parameters. Welding with micro-jet cooling could be treated as a very promising welding Docol steels process with high industrial application. Results of non-destructive inspections on macro samples corresponded with further destructive test results (tensile strength, hardness, fatigue, metallographic structure analyses). This article aims to verify fatigue behavior of Docol 1200 M steel after welding supported by the cooling using the micro-jet technique. For the first time, micro-jet cooling was used to weld this kind of steel to check the mechanical properties of the joint, especially to determine the fatigue limit. This study is formulated as follows: investigating fatigue resistance of the Docol 1200 M weld manufactured at the cooling process with micro-jets. The joints were produced in the MAG (Metal Active Gas) technology modified by micro-jet cooling. The results collected in the fatigue test were processed in the form of the Wöhler’s S–N diagram following the fatigue limit of the weld examined. All data have indicated the possibility of obtaining a new method of welded joints with high fatigue limit minimum of 480 MPa. It could be important to achieve a tensile strength of 700 MPa while maintaining the best relative elongation at the level of the base material.

## 1. Introduction

The progress in the automotive industry is created by the application of modern materials and types of joining as well as new components and constructions [1,2,3,4]. If all science-engineering fields are reached and collated, then a final product is manufactured in the right quality without a waste of time for its prototyping, obtaining beneficial features such as a 90 kg reduction of the nominal mass of the vehicle and 5 kg lowering of Body in White (BIW) in comparison to the previous generation [1]. It is possible by employing beneficial materials such as Advanced High Strength Steel (AHSS) and Ultra High Strength Steel (UHSS), which possess very attractive mechanical parameters [3,5,6,7]. Therefore, this kind of steel is used not only in typical cars with an internal combustion engine as well as in Hybrid Electric Vehicles (HEV) increasing stiffness by 12% and 10% for bending and torsion loading, respectively [8]. Moreover, it is taken for producing an electric car battery case, being a component of BIW for battery replacement and safety regimes. In this case, the martensitic steel series of Docol 1700 M plays an important role because of high mechanical properties as a result of 3D manufacturing technology. This has been employed since the year 2020 in the Ford Explorer, indicating its actual application [1]. Moreover, this type of steel is employed for producing sill and roof reinforcements as well as door/side intrusion beams, bumpers, and seat structures [9]. With respect to different components and joining processes used for manufacturing of the elements made of this kind of material, and automotive industry regimes for quality and mechanical resistance of this type of components, the steel requires more attention, taking chemical composition, mechanical properties, behavior under fatigue, as well as features due to welding. Joining Docol steels with classic welding methods allows to obtain correct welded joints, but with a much lower fatigue strength than that of the parent material. The aim of the article is to discuss a newly developed welding with micro-jet cooling stage [10,11], ensuring the production of repeatable welded joints with high fatigue properties. The newly developed process made it possible to change the microstructure of the joint, which resulted in an increase in its strength. The aim of this article is to extend the knowledge on this kind of steel and the welding process connecting with micro-jet cooling (Figure 1), and the following sections cover all mentioned material and process aspects.

Docol Advanced High-Strength Steel (AHSS) are obtained in the technological process such as: hot and cold rolled as well as hot-dip and electro galvanized. That material range includes thin sheet steel with a thickness range from 0.4 mm to 16 mm. Docol steels have become particularly attractive in the automotive industry for three important reasons [1,4,12]:high ultimate tensile strength, up to 1700 MPa,high yield stress, up to 1350 MPa,elongation, up to 6%.

The application of Docol steels in the automotive branch of industry results from reducing the thickness of the car body at also acceptable or more attractive mechanical parameters in comparison to typical structural materials. For the manufacturing of vehicles, attempts are being made to use high-strength martensitic steels from the Docol group [13,14,15,16]. They make it possible to reduce the weight of the vehicle and reduce energy consumption for its production [1,9].

For instance, Reference [17] presents the results of Docol 1200 M martensitic steel welding. In this study, the very thick sheets were joined in a lap joint configuration. The influence of various welding parameters (especially linear heat input) during welding on structural changes was examined by the occurrence mainly of dominant phases: martensite and fine ferrite. Reference [18] presents the weldability of three different modern high-strength steel plates, with a thickness of 6 and 8 mm. Two-passes Metal Active Gas (MAG) welding process was used with various heat inputs. In this case as well, the test results revealed that increasing the heat input resulted in lower tensile strength of the joint. However, the authors of Reference [19] are of a different opinion. The authors prepared a mixed joint of high-strength steels: S460N and S460ML. The joints were manufactured using various values of heat input for each welding bead. The results of prepared welds showed that using higher heat input has a significant influence on the mechanical properties of the mixed steel joints, as presented in [20].

In Reference [18] the authors have shown that high-strength low alloy (HSLA) steels are quite crucial in weight reduction in the manufacturing of vehicles and constructions with high steel thickness. Quenched and tempered S690QL HSLA steel with 700 MPa yield stress was multi-pass welded with seven passes using MAG (Metal Active Gas) and FCAW (Flux Cored Arc Welding) methods. The effects of the process parameters for both methods on weld metals and heat affected zones (HAZ) were investigated. The authors suggested that welding with higher heat input influenced the mechanical properties.

In summary, it can be concluded that the modern high-strength steel plates have an excellent combination of strength, fracture toughness, and resistance on impact based on micro-alloying and complex microstructure [21]. Retaining this combination of properties in the weld zone is a major challenge for applications in high-demanding structural construction. The problem that has not been solved so far is the development of a repeatable technology of joining these steels, ensuring obtaining joints with sufficiently ultimate tensile strength at the level of 600 MPa and higher without welding defects. With these steels, the plastic properties of the joint were always worse than that of the base material, as evidenced by different values of the relative elongation.

This research aimed to investigate the mechanical properties of the high-strength steel Docol 1200 M after MAG welding with micro-jet cooling. The utilitarian goal of the study is to develop a welding method for these steels. The new technology will be based on classic MIG and TIG (Metal Inert Gas and Tungsten Inert Gas) processes, modernized through the use of cooling attachments, affecting the structure of the connections obtained. It was assumed that the selection of welding and micro-second cooling parameters would allow for crack-free joints with a significant amount of fine-grained martensite, which translates into high-strength properties of the welds.

The newly developed technology for the analyzed steels will allow its use in the automotive industry and should contribute to a wider use of the discussed steels in practice.

These steels are relatively not very well weldable because of the cracks can occur in the weld and Heat Affected Zone [2,22,23,24]. This is related to carbon content and its equivalent [21]. The CE (Carbon Equivalent) coefficient is very similar to that of low-alloy steels [25,26]. An important disadvantage is that the mechanical parameter of the welded Docol joint is much lower than that of the parent material. For example, the tensile strength of Docol 1200 M steel is on the level of 1200 MPa, whereas the joint strength is nearly half times lower and equals not more than 700 MPa. In this paper, we analyze for the first time the possibilities of AHSS steel welding with micro-jet cooling [7,8,27,28]. Micro-jet cooling enables to control the microstructure of the welds manufactured. The considered method has been applied efficiently in joining low-alloy steels as well as aluminum alloys [2].

Selecting the steel type to the application is conducted based on a range of the material type from Docol 900 M up to Docol 1500 M at details of chemical composition [29] and mechanical properties (Figure 1). As can be noticed in Table 1, the relationship between the C and Mn elements follows the steel type, reaching the value of 0.03 at Docol 900 M and 0.19 at Docol 1500 M. From Figure 2a,b it is easy to note that the yield stress and ultimate tensile strength of Docol steels is represented by the wide range of values of 700 MPa to 1350 MPa and respectively 900 MPa to 1700 MPa at ductility min. 3%. The similar results were presented in [29]. With respect to an engineering point of view, a relationship between the mechanical parameters (Figure 2b) is important to assess the quality of the steel and its welding because the proportion enables to indicate a size of hardening region at the stress that reaches the yield point up to ultimate tensile strength. In this approach, an elongation plays a significant role because the parameter is directly related to ductility, enabling to avoid brittle cracking.

During AHSS steels welding, the regime determined by thermodynamic analysis of the process should be reached. The linear energy up to 4 kJ/cm is required to refine the ferrite and reduce stress due to welding process [23,24,25]. Preheating and control the temperature of the interpass layer is needed for lowering the hydrogen content in the joint. It can be explained as follows: single hydrogen atoms H are created in the weld, which can freely permeate between iron atoms, where they combine into the H_2_ molecule (recombination). Hydrogen accumulation in the metal leads to internal pressure, causing an increase of stress, which in turn causes Hydrogen Induced Cracking (HIC) [26,30,31]. HIC in Docol steels appears mainly at the martensite-ferrite grain boundaries as well as at interaction with non-metallic inclusions [2,7]. Hydrogen cracking usually spreads parallel to the surface of the sheet [1,6,8]. The results of Docol steels welding, presented in literature, show that the welding joint does not crack due to stress corrosion cracking (SSCC—Sulfide Stress Corrosion Cracking), because these steels contain traces of sulfur [27]. When welding thin-walled structures, attempts have been made to weld it with varying degrees of success without preheating [2]. The investigation results presented in [2] confirm that the obtained data are not reproducible. In order to manufacture the weld at the attractive technical quality, it was decided to weld Docol 1200 M steel without preheating and micro-jet cooling. Joints with different MAG welding parameters and different parameters of cooling micro-jet were made. When welding various types of steel (ferritic, austenitic, martensitic), it is very important to select the right shielding gas and wire [32].

As can be noticed in [20,22,23], fatigue and tensile experiments are crucial destructive experiments in the assessment of AHSS welds quality because they follow important mechanical parameters used in engineering practice and research efforts. Moreover, this kind of data is significantly expected by vehicle manufacturers. Steel Docol 1200 M joints were not manufactured applying micro-jet cooling process. It is important to capture the fatigue strength because this parameter is the important data for assessment on the quality of the weld [16,31,33]. Therefore, the paper follows the weld behavior under static and fatigue testing.

## 2. Materials and Methods

A Docol 1200 M welded (BW) butt joints at a thickness of 2 mm was manufactured. The requirements of EN 15614-1 standard were used for the MAG welding method at the low position (PA). The single-stitch welding and a finished 3 mm thick weld are illustrated in Figure 3. The sample size for testing the weldability of Docol 1200 M (HR-Hot Rolled) steel was accomplished according to Figure 3. Sample total dimensions: 3 mm × 200 mm × 400 mm.

It was decided to produce welds at MAG (Metal Active Gas) process using the gas mixture as of: Ar + 18% CO_2_ (the PN-EN 14,175 standard). All of the samples were welded with an electrode wire EN ISO 16834-A G 89 6 M21 Mn4Ni2CrMo−UNION X90 (Böhler Düsseldor, Germany) at the chemical composition presented in [4]. It is easy to note that UNION X90 wire contained Cr and Mo with the amount of providing good plastic properties [4]. The detailed welding process parameters for sheets with a thickness of 3 mm are provided in Table 1.

### Methods, Scope of Research

The scope of research included non-destructive testing (NDT):Visual tests (VT) of welded joints were conducted with the eye fitted with a magnifying glass at 3× magnification—the test was performed according to the requirements of PN-EN ISO 17,638 standard and the assessment criteria according to EN ISO 5817 standard,Magnetic-powder tests (MT) were carried out in accordance with the PN-EN ISO 17,638 standard and the assessment criteria according to EN ISO 5817 standard using a magnetic flaw detector device type REM-230 (ATG, Prague, Czech Republic).

Evaluation of microstructure and mechanical properties:Examinations of the microstructure of the samples digested with the Adler reagent using a light microscope (LM),Hardness measurement was done focusing on guidelines of the PN-EN ISO 9015-1:2011 [34] and PN-EN ISO 6507-1:2018-05 [35] standards,Tensile test was carried out basing on the regimes of the PN-EN ISO 6892-1:2020 standard [36]. The experimental approach was elaborated design to follow the Docol 1200 M weld to Docol 1200 M behavior under static and fatigue loading. Therefore, the U-notched specimen was designed. It was done basing on requirements of the E 468–90 ASTM standard [37] for fatigue tests,Fatigue test collecting specimen designing was performed at rules of the ASTM E468-18 standard [38].

## 3. Results and Discussion

The article presents the possibility of welding a thin-walled structure (3 mm) made of high-strength Docol 1200 M steel. These steels are difficult to weld [39,40], as cracks often occur. The big disadvantage is obtaining much worse strength and elongation of the welded joint in relation to the base material. For this purpose, it was decided to use micro-jet cooling for welding. This section presents selected test results and their analysis.

To further enhance the mechanical properties of joints after MAG welding, an addition of micro-jet cooling technology was selected. For the steel structure welding process, the following micro-jet cooling parameters were implemented:quantity of cooling nozzles: 1,form of cooling medium: Ar or He,pressure of the cooling medium: 0.6−0.7 MPa,diameter of micro stream: 60−70 µm,distance of the micro-jet nozzle from the welded surface: 20 mm.

After the welding with micro-jet cooling, the following non-destructive tests (NDT) were carried out: visual (VT), magnetic-particle (MT), and radiographic. The gap between the welded elements arm-gap (Figure 2) was checked in a range of 1 to 2 mm, 0.5 mm step by various parameters of micro-jet cooling after MAG welding. A correctly selected gap between elements together with micro-jet cooling parameters affect the cooling conditions of the weld. The results of the welding process are presented in Table 2. The comparison of the results of non-destructive tests (Table 2) confirms that all connections made without micro-jet cooling have welding defects. As it was investigated, the micro-jet cooling has allowed us to manufacture joints that did not show any welding defects or incompatibilities in the tests performed.

Results of non-destructive tests exhibited no cracks occurrence in the weld manufactured at the micro-jet cooling technique (Figure 4).

The table data shows that the gap between elements should be equal to 1.5 mm and micro-jet cooling should not be too intensive (samples S1.5a, S1.5b, S1.5c). For the 2 mm gap, no cracks were observed, only for low-intensity helium micro-jet cooling (sample S2b) (Table 2). It was easy to assume that helium micro-jet cooling gives better results. Further joint hardness distribution was also carried out. Samples with positive results from non-destructive tests (NDT) were only tested (the gap between elements was only 1.5 mm, micro-jet cooling was always not intensive). Two micro-jet gases were used, of which micro-jet cooling helium gave better results (samples S1.5a, S1.5b, S1.5c, S1.5e), which was confirmed by non-destructive tests. Analyzing the data from Table 3, it can be noted that the micro-jet cooling should be used to the welding process. Micro-jet cooling parameters do not have a special effect on the hardness values (Table 3), enabling to create the mechanical parameter of the weld and base metal on the same level.

After the hardness assessment, the strength tests of the welded elements were carried out and the 3369 INSTRON testing machine (INSTRON, High Wycombe, England) was selected for that purpose. Two analyses were performed:strength of Docol 1200 M steel with the use of argon micro-jet cooling,strength of Docol 1200 M steel with the use of helium micro-jet cooling.

Micro-jet gas pressure (Ar or He) was equal to 0.7 MPa, micro-jet stream diameter was 60 µm. Each test was repeated 3 times (samples 1, 2, 3). Tensile strength and fatigue tests were performed as the main research to check the quality of the joint. The results are presented in Table 4.

As captured in the tensile test, the Docol 1200 M steel is a very attractive structural material because of the value of proportional limit taking 771 MPa, ultimate tensile strength exceeding 1200 MPa, and elongation that reaches min. 8% (Figure 5a). These values protect the components made of this kind of steel under operational conditions against deformation and fatigue damages. Taking the fracture zone of the Docol 1200 M steel, the stress state components playing a role in the material cracking under tension can be indicated, i.e., axial and shear. The steel and weld were also examined under the micrographic structure (Figure 6) and tensile test applying hourglass specimen in the geometry and dimensions shown in Figure 7a. It was assessed based on tensile characteristics determined on the flat and hourglass specimens (Figure 7b). The specimen was adopted (Figure 7) to the conducted tests by allocating the weld in the middle of the measuring zone. The comparison of results from the tensile tests indicates an increase of strength parameters at the reduction of elongation as an effect of a stress concentration. In the case of the micro-jet cooling weld of the Docol 1200 M, the same effect was noticed, but the lowering of the elongation value was not significant (Figure 5). The role of the weld joining in the steel behavior under tension is determined by a comparison of the tensile characteristic of the parent material and the weld (Figure 5). It was visible in the 50% lowering of proportional limit, yield stress, and ultimate tensile strength at almost the same elongation.

Application of micro-jet cooling during the welding process improved YS and UTS values. The average YS value of MAG welded joint without micro-jet cooling was 495 MPa and the average UTS value was 723 MPa. They increased respectively to the values 551 MPa (YS) and 751 MPa (UTS) for argon micro-jet cooling, and respectively to the values 534 MPa (YS) and 762 MPa (UTS) for helium micro-jet cooling. It has been confirmed from NDT observations that helium micro-jet cooling gives better results. Next, the microstructure analysis was carried out. A typical microstructure of weld is presented in Figure 6. This figure illustrates the microstructure of the cross-section of the joint, where the martensitic component is clearly visible. Welds are free from defects and incompatibilities. Finally, fatigue tests were carried out. Only the welding process with helium micro-jet cooling was analyzed. Micro-jet gas pressure (He only) was equal to 0.7 MPa, micro-jet stream diameter was 60 µm.

Fatigue tests were employed to follow the behavior of the micro-cooling weld joining of the Docol 1200 M under cyclic loading [41,42]. All tests were carried out at room temperature using the 8874 INSTRON servo-hydraulic testing machine (INSTRON, High Wycombe, England) and stress signal having the following parameters: stress ratio = 0, the range of the maximum values of stress from 500 MPa up to 700 MPa, frequency = 5 Hz (Figure 8). The stress signals were in the form of a cyclic function having a sinusoidal shape (Figure 9). The last 50 cycles of the fatigue tests were taken for capturing the weld behavior before fracture appearing. Practically, they were collected in the other files directly related to the analysis taken. The hourglass specimens having the weld in the middle section of the measuring region were used (Figure 7). The specimens were selected with the Docol 1200 M steel in a form of the sheet using a water cutting technology.

The fatigue tests have reflected the weld response on cyclic loading up to fracture (Figure 8 and Figure 11). As can be noticed on the fracture zone from the test at the maximum values of stress of 500 MPa (Figure 10) and 700 MPa (Figure 11), the fatigue damages occurred in the middle of the specimen, following directly the degradation of the micro-jet cooling weld under the cyclic loading. Differences in the weld behavior at various values of stress were expressed by changes in the orientation of the fracture regions, i.e., with an increase of the stress value, the axial stress was more dominant than the shear stress, showing a gradual vanishing of the second-mentioned stress component and it disappearing at the highest value of the stress applied (Figure 11). Moreover, variations in displacement versus a number of cycles, employed to follow the weld response on the cyclic loading, were also clearly visible. They were represented by the reduction of non-linear course of the relationship with the increase of stress value directly before the weld fracture.

At 500 MPa and 550 MPa, the region degradation occurred at relatively little plastic deformation (Figure 12a,b), compared to the results for the bigger values i.e., 650 MPa and 750 MPa (Figure 12c,d). This kind of data can be very easily connected with results from tensile tests (Figure 5b). As it is possible to notice, the features of the zones captured are dependent on the relationship of values of the cyclic stress to mechanical parameters. If a value of the cyclic stress is close to the ultimate tensile strength, then the plastic deformation is the dominant mechanism in the weld degradation.

Fatigue of the micro-jet cooling weld is also represented by the fracture at the following maximum values of stress: 700 MPa, 650 MPa, 600 MPa, 550 MPa, and 500 MPa, taking the relationship presented in Figure 13. From the engineering point of view, these results have enabled to predict fatigue durability of the weld tested, at cyclic stress within the range from yield stress up to ultimate tensile strength, as well as determine the fatigue limit. It has enabled to indicate the value of stress to operational conditions, which does not lead to fatigue damages initiation. In the case of the weld from the technological process supported by helium micro-jet cooling, the stress reached 480 MPa.

The obtained results show that:Two micro-jet gases were used, of which micro-jet cooling helium gave better results (samples S1.5a, S1.5b, S1.5c, S1.5e), which was confirmed by non-destructive tests (Table 2, Figure 4).Tensile strength and fatigue tests were performed as the main research to check the quality of the joint. It was noticed that the joints with better plastic properties were obtained when micro-jet cooling was additionally used for MAG welding. Relative elongation increased from 8% to 11% when micro-jet cooling was used (Table 4, Figure 5).The yield stress of the joint without micro-jet cooling was below the required value of 500 MPa, while welding with micro-jet cooling resulted in a yield stress value of 520 MPa both when helium and argon were used in micro-jet cooling. Similarly, a higher tensile strength was obtained thanks to the use of micro-jet cooling.Yield stress and tensile strength values were at a similar level of 750 MPa when helium or argon was used for micro-jet cooling. The joint after welding with He micro-jet cooling had slightly higher elongation (11%) in comparison with Ar micro-jet cooling (10%); therefore, helium was selected as the micro-jet gas for fatigue tests. Without micro-jet cooling, tensile strength was obtained at a lower level of 720 MPa (Table 4).

Generally, the fatigue tests confirmed that the joints are of high quality. The Wöhler curve has been established as a key characteristic for determining the quality of the weld examined. The fatigue strength was obtained at a satisfactory level of 480 MPa (Figure 10). Therefore, this kind of data can be used by a lot of engineer and research groups for designing and modeling. It is worth noting that data in the form collected are presented purely indicating an insufficient number of results for designing the high-strength steel behavior. Non-destructive and destructive tests confirmed that the correct joint was obtained from Docol 1200 M steel thanks to the use of micro-jet cooling in MAG welding.

## 4. Summary 

The welded joints were made of the difficult-to-weld material Docol 1200 M. The most favorable welding conditions and parameters were determined. The possibility of welding Docol MAG steel with micro-jet cooling has been tested. The most important parameters of micro-jet cooling have been thoroughly checked. The most suitable micro-jet gas was selected. Non-destructive and destructive tests were carried out to assess the quality of the joint. The microscope (LM) observations, tensile and hardness tests, as well as fatigue approaches were carried out. Fatigue tests were treated as the most important destructive experiment in the assessment of the quality of Docol welds. The results from the tensile test indicated that the Docol 1200M steel is sensitive to the micro-jet welding processes, expressing half-smaller values of mechanical parameters than the parent material, not including elongation. Independently of the material state examined, the value of this parameter was very similar. The weld behavior under fatigue had a range of significant differences in the number of cycles to fracture at the stress value from 700 MPa up to the fatigue limit of 480 MPa. This was also reflected by a gradual fracturing of the steel at the smaller values of stress.

## 5. Conclusions

Results from non-destructive and destructive tests on the weld joint manufactured at the micro-jet cooling have enabled us to formulate the following concluding remarks:the micro-jet cooling technique allows producing the weld for Docol 1200 M steel with a martensitic microstructure similar to the parent material;mechanical properties of the weld supported by the micro-jet cooling have expressed 50% lowering and maintaining elongation value in comparison to the steel in the as-received state;tensile curve of the weld with respect to the welding process used has received more regular shape, which is requested by engineers for designing and research for modeling;fatigue limit of the weld tested was very beneficial because of the 480 MPa that was determined. This value can be directly employed in designing, calculations, and modeling as fundamental data for assessing the technical state of a component;the results have confirmed that the elaborated joining technology allows obtaining the assumed parameters of the joint with respect to high plastic properties (elongation = 10%), comparable to the parent material. Publications indicate that, so far, high and repeatable parameters of the joint have not yet been obtained without micro-jet cooling.

## Figures and Tables

**Figure 1 materials-14-00936-f001:**
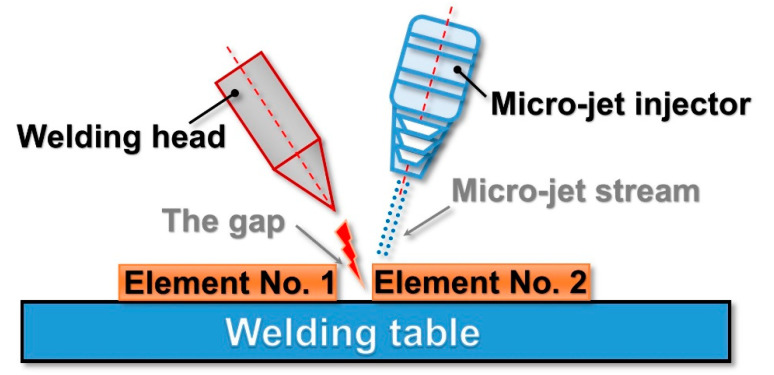
A scheme of Metal Active Gas (MAG) welding with micro-jet technique.

**Figure 2 materials-14-00936-f002:**
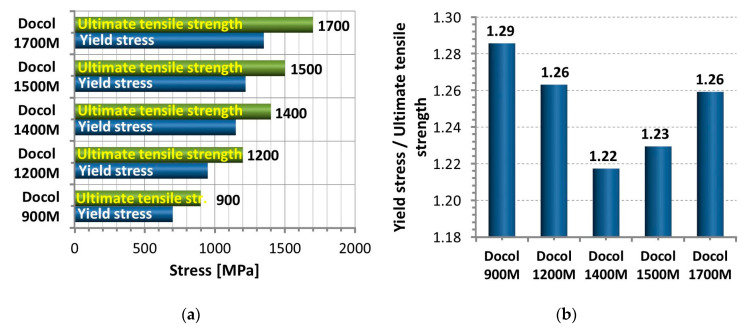
Yield stress and ultimate tensile strength of different types of Docol steel (**a**) on the basis of selected data from the Product Program SSAB, (**b**) the results of own analysis.

**Figure 3 materials-14-00936-f003:**
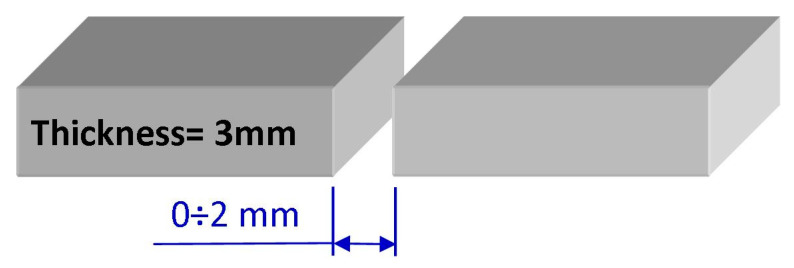
A scheme for preparing elements for MAG welding supported by the micro-jet cooling method, thickness t = 2 mm.

**Figure 4 materials-14-00936-f004:**
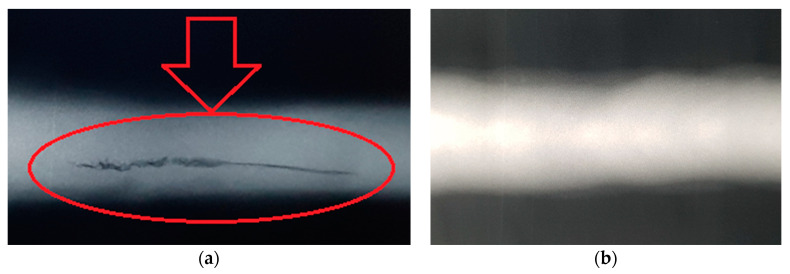
Weld after welding without (**a**) and with (**b**) micro-jet cooling in X-ray inspection.

**Figure 5 materials-14-00936-f005:**
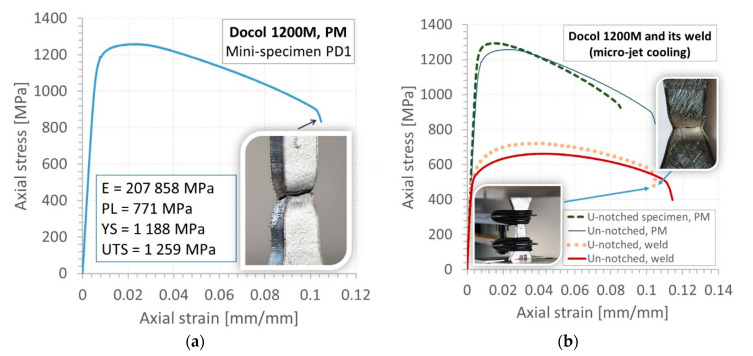
Tensile characteristics of the Docol 1200 M: (**a**) flat (un-notched) mini-specimen (**b**) applying U-notched and un-notched specimens to parent material (PM) and weld (micro-jet), E—Young’s modulus, PL—proportional limit, YS—yield stress, UTS—ultimate tensile strength.

**Figure 6 materials-14-00936-f006:**
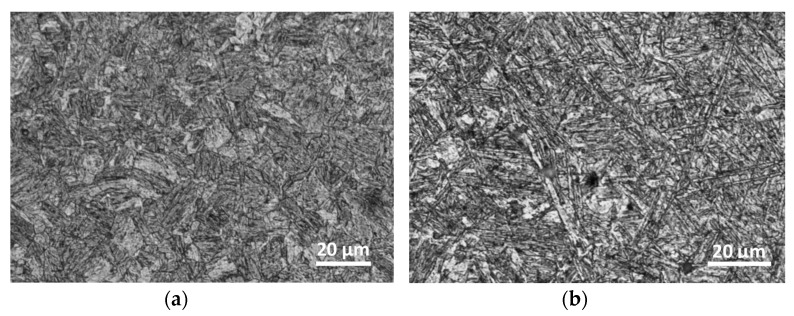
Microstructure of the joint cross-section without (**a**) and with the use of helium micro-jet cooling (**b**), etched Nital, time 12 s.

**Figure 7 materials-14-00936-f007:**
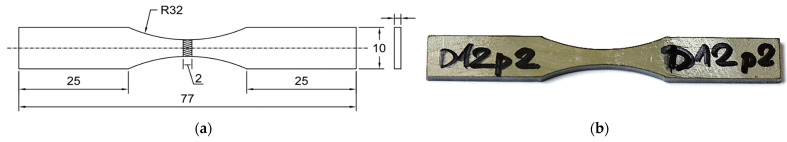
The hourglass specimen with the weld: (**a**) geometry, dimensions, and the weld location (unit: mm), (**b**) manufactured specimen, the nominal thickness was equal to 1.8 mm.

**Figure 8 materials-14-00936-f008:**
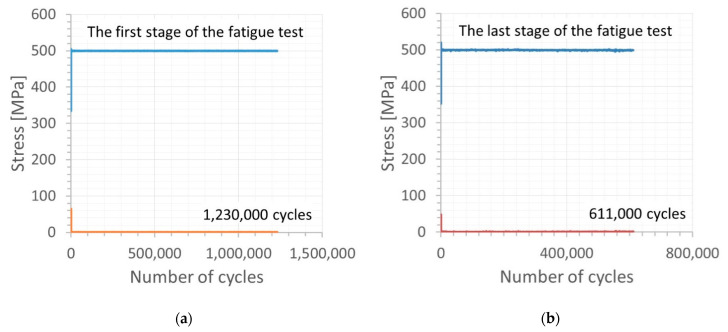
Maximum and minimum values of stress signal for the cycles: (**a**) and (**b**) up to 1,230,000, and 1,841,000 (sum before the material fracture = 1,841,757 cycles), respectively.

**Figure 9 materials-14-00936-f009:**
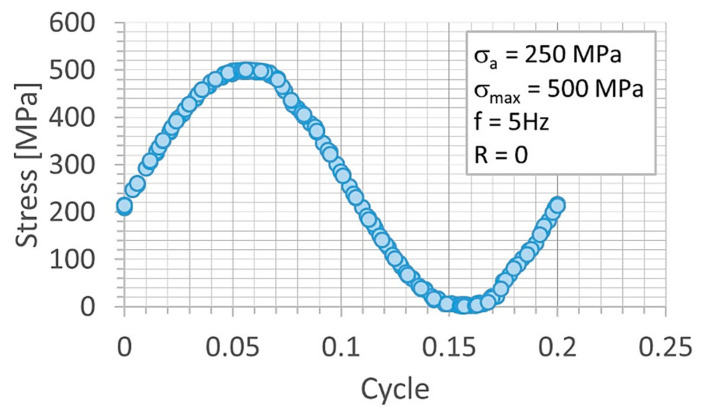
Stress signal at amplitude of 250 MPa and maximum value of 500 MPa, frequency 5 Hz.

**Figure 10 materials-14-00936-f010:**
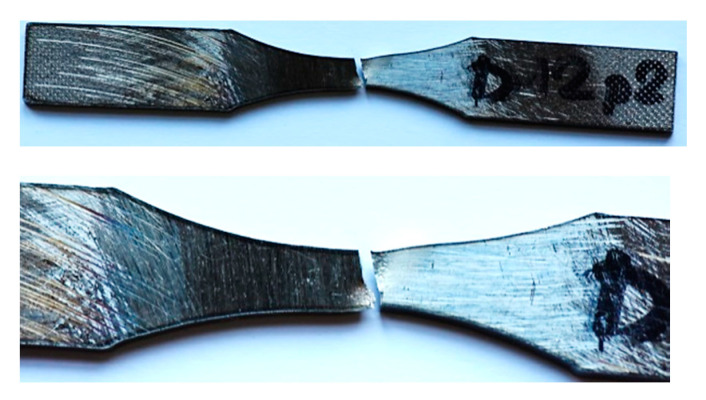
The hourglass specimen made of the Docol 1200 M steel with the micro-jet cooling weld after fatigue test at maximum value of stress of 500 MPa.

**Figure 11 materials-14-00936-f011:**
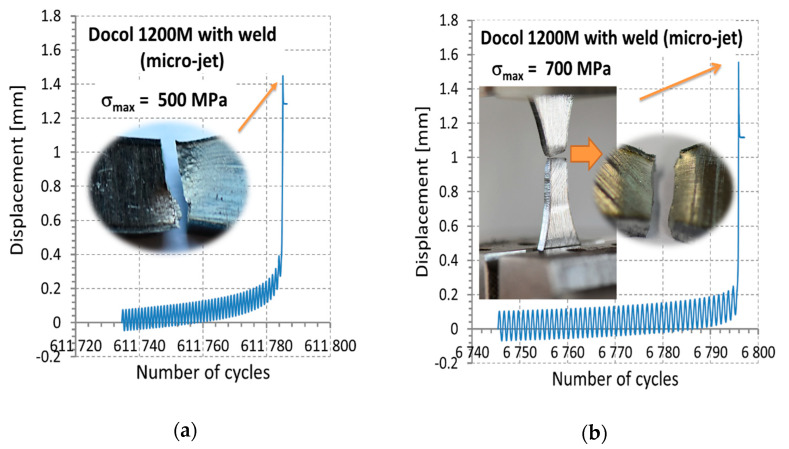
The last stage of fatigue tests before fracture of the Docol 1200 M steel’s micro-jet cooling weld for the following maximum values of stress: (**a**) 500 MPa and (**b**) 700 MPa.

**Figure 12 materials-14-00936-f012:**
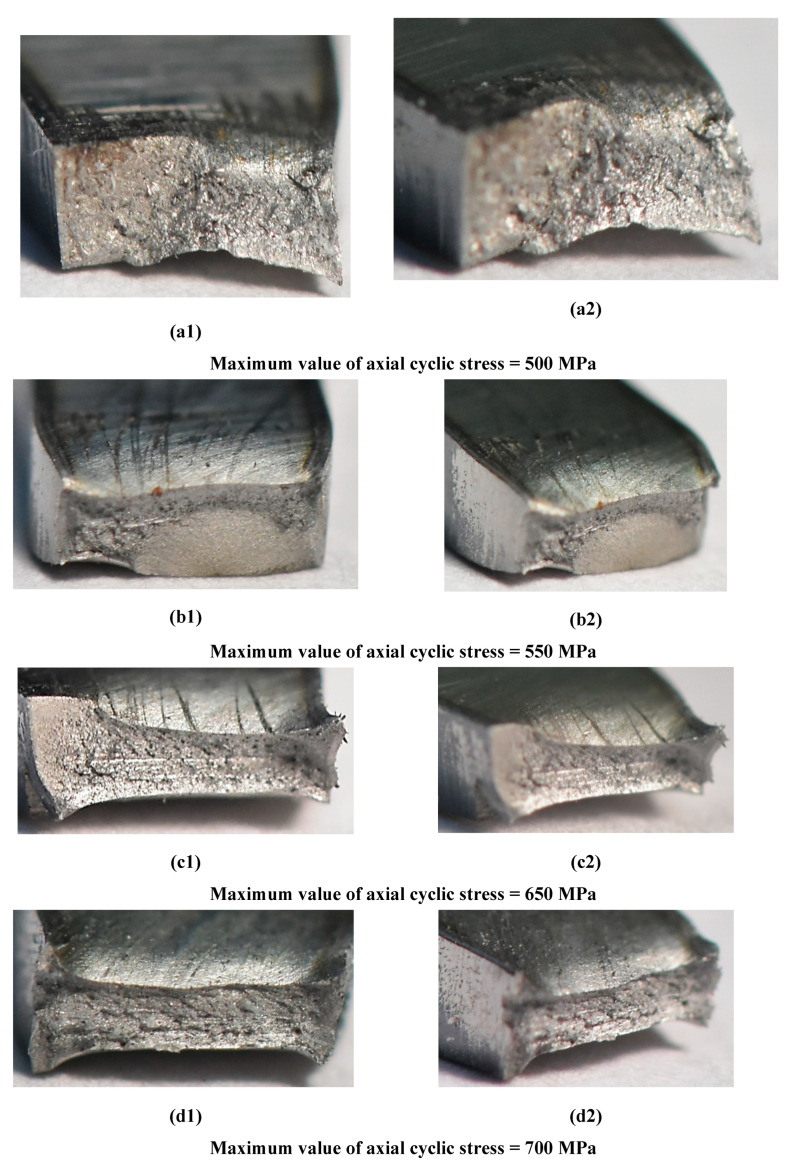
Fracture zones of the Docol 1200 M weld manufactured at the micro-jet cooling technique after fatigue test under the maximum values of stress of 500 MPa to 700 MPa.

**Figure 13 materials-14-00936-f013:**
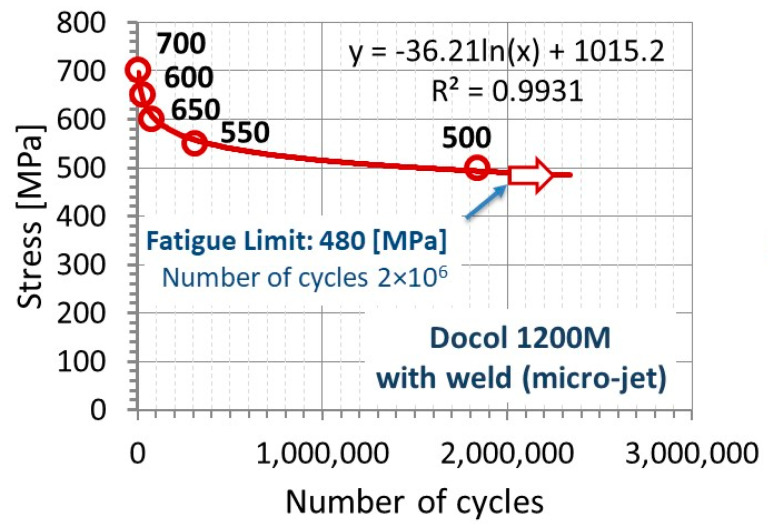
The Wöhler curve of the weld with micro-jet cooling technology on the Docol 1200 M steel.

**Table 1 materials-14-00936-t001:** Sample welding details.

Order of Layers	Method of Welding	Diameter of Wire (mm)	Current (A)	Voltage (V)	Polarization	Welding Speed (mm/min)	Energy (kJ/cm)
1	MAG	1.0	109	19	DC “+”	350	3.3

**Table 2 materials-14-00936-t002:** Results from non-destructive investigations, the joint application-movable platform.

Micro-Jet Stream Diameter (μm)	Micro-Jet Stream Pressure (MPa)	Gap (mm)	Micro-Jet Gas	Symbol of Sample	Observation
without	without	1	without	S1o	Cracks in the weld
60	0.6	1	Ar or He	S1a	Cracks in the weld
60	0.7	1	Ar or He	S1b	Cracks in the weld
70	0.6	1	Ar or He	S1c	Cracks in the weld
70	0.7	1	Ar or He	S1d	Cracks in the weld
without	without	1.5	Ar or He	S1.5o	Cracks in the weld
60	0.6	1.5	Ar or He	S1.5a	No cracks
60	0.7	1.5	Ar or He	S1.5b	No cracks
70	0.6	1.5	Ar or He	S1.5c	No cracks
70	0.7	1.5	Ar	S1.5d	Cracks in the weld
70	0.7	1.5	He	S1.5e	No cracks
without	without	2	without	S2o	Cracks in the weld
60	0.6	2	Ar	S2a	Cracks in the weld
60	0.6	2	He	S2b	No cracks
60	0.7	2	Ar or He	S2c	Cracks in the weld
70	0.6	2	Ar or He	S2d	Cracks in the weld
70	0.7	2	Ar or He	S2e	Cracks in the weld

**Table 3 materials-14-00936-t003:** Hardness values in a mixed joint at selected parameters of welding process, HV—Vickers Hardness.

Micro-Jet Stream Diameter (µm)	Micro-Jet Stream Pressure (MPa)	Micro-Jet Gas	Parent Material HV (MPa)	HAZ HV (MPa)	Weld HV (MPa)
60	0.6	Ar	335	359	341
60	0.7	Ar	336	352	343
60	0.6	Ar	334	351	341
Average value of HV			335	354	342
Standard deviation of HV			±0.8	±3.6	±0.9
60	0.7	He	333	350	340
70	0.6	He	335	349	341
60	0.6	He	334	347	339
Average value of HV			334	349	340
Standard deviation of HV			±0.8	±1.2	±0.8
without	without	without	334	365	351

**Table 4 materials-14-00936-t004:** Tensile tests results of welded Docol 1200 M steel without the use of micro-jet cooling ± measurement uncertainty.

Micro-Jet Gas	Yield Stress, YS (MPa)	Ultimate Tensile Strength, UTS, (MPa)	A_5min_ (%)
without	495 ± 8	723 ± 10	8
Ar	521 ± 12	751 ± 14	10 11
He	534 ± 10	762 ± 12	

## Data Availability

Data sharing is not applicable to this article.

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
