# Peer review of "High Martensitic Steel after Welding with Micro-Jet Cooling in Microstructural and Mechanical Investigations"

_materials, 2021, doi:10.3390/ma14040936_

Round 1

Reviewer 1 Report

The reviewed article deals with high martensitic steel welded with micro-jet cooling and microstructural and mechanical investigations of joints. The list of detailed remarks is given below:

  1. Abstract - last sentence should be translate in English.
  2. Introduction - this part is huge, please make this chapter shorter, leaving the essence of it.
  3. Table 1 - please add "wt%". Moreover, if there is 0.15 Nb + Ti and 0.12% Ti, so Nb is 0.03? please explain it.
  4. Figure 1: (a) - it is repetition data from Table 2, so please carefully think about which form of data presentation is better and select it; (b) - it could be packed in Table 2.
  5. Please add dimensions of plates except thickness.
  6. Table 3 - please add reference.
  7. Table 4 - current instead amperage.
  8. VT standard is ISO 17637. Moreover, please use standard instead norm.
  9. Why sometimes standards are like a reference and sometmes no, please explain it.
  10. Hardness measurements instead probe.
  11. Table 5 - please add "sample code" in order to clearly indication of the sample.
  12. Scheme of welding set-up should be added.
  13. Table 6 - why it is in MPa? what was the maximum load during hardness measurement? it was only one measurement in one zone for each sample? Please add average values with standard deviation.
  14. "Analysing the data from the Table 6 it can be noted that micro-jet cooling must be used to welding process. Micro-jet cooling parameters do not have special effect on the hardness values." - please explain it, I do not understand this conclusion (below Table 6).
  15. "Micro-jet gas pressure (Ar or He) was equal 70 MPa" - such high pressure? in Table 5 and Table 6 it is 0.6 and 0.7 MPa.
  16. Table 7 - so if test was repeated 3 times, so it should be average value and standard deviation, for all data. Please complete it.
  17. Figure 4 - please add new scale bar, this is invisible.
  18. There is no discussion with literature, please complete it. Moreover, in the title it is "microstructural investigation", while in text it is only one image. There is no fractured surfaces after tests.

I have found a plagiarism in reviewed article. It concern Figure 4, which I found in article below (Figure 3): "Welding of DOCOL 1200M using micro-jet cooling". DOI: https://doi.org/10.26628/wtr.v92i1.1087

Author Response

Thank you very much for you profesional and kind suggestions

Reviewer 2 Report

The manuscript is devoted to study the effect of micro-jet cooling on microstructural and mechanical properties of high martensitic steel welded joints. Although authors presented some experimental results, the results are not support the arguments. Therefore, this manuscript in this form is not suitable to publish in Materials unless authors show more results and address the following comments.

1- Arrangment of manuscript is not acceptable. There is no “ Results and discussion” section! 

2- Abstract: what is “h” in the sentence “ This was confirmed in h tensile and hardness tests …”

3- Methodes, scope of research: “Visual test on macro samples…” Visual inspection is not a destructive test, it is non-destructive test then remove it from this part.

4- Authors displayed the visual exmaniation on the appearance of  the welds in table 5 with some crakcs and sound welds without showing any evidence. Therefore it is required to add some images of weld appearance at leat one sound weld and one cracked weld.

5- It has been claimed that He micro-jet cooling exhibits the best results. However there is no significant differenece between mechanical propertis of without and Ar micro-jet cooling welds.

6- To underestnad the effect of micro-jet cooling with different parameters, authors would better to show the microstructure of unwelded parent metal, without, Ar, He micro-jet cooling welds all together and include the them in Fig. 4 (scale also is not clear in this figure). 

Author Response

(The authors gave the same response as above.)

Reviewer 3 Report

The manuscript "High Martensitic Steel after Welding with Micro-Jet Cooling in Microstructural and Mechanical Investigations" has been reviewed.

It deals with an experimental investigation on the fatigue life of welded steel joints.

It is an interesting work but english should be deeply revised by a native speaker. Some sentences require grammar check despite the sense, by the way, more or less is clear. 

Some major revisions are required. In the following the details:

In my opinion the title should be modified to be more representative of the work.

Abstract. Remove last three lines at the end of the abstract or translate from polish to english.

Introduction. Line 5. add a space between 90 and kg, add a space between 5 and kg.

Pag. 2, 6 lines from the end. ...acicular, and allotriomorphic ferrite or martensite. Please add a reference or evidence for that!

Pag. 3. Third paragraph. In the article []. Which one?

Pag. 3. 4th paragraph "Przy tych stali zawsze uzyskiwano własności plastyczne złacza gorsze od materiału rodzimego, czego dowodem są różne wartości wydłużenia względnego"??? Please translate in english!

Fig. 1. Yield stress/ultimate stress is wrong. It is right ultimate stress/Yield stress. Please check.

Tables 1 and 3. Composition wt% or at%?

Table 3. Please add the motivation for the selection of this wire.

Table 2. Thermal curing??? What does it mean? It's a steel, not a polymer. Post welding heat treatment (PWHT)??? If yes, please indicate temperature and time.

Table 4 welding method 135. Please add details.

Table 4. Volatage. Please check.

Table 6. In the title is reported hardness value, in the table MPa. Which is right? In case of HV indicate the applied load (as required from the standard).

Fig. 2. Please indicate the source.

Pag. 6 2nd paragraph. Hardness, not harness.

Table 6 in the colum A% only 2 values are reported (instead of 3...). Check.

Fig. 4 microstructure. Please indicate details about ethants and time.

Conclusion line 3 jet not jest

Conclusion line 12 This value this kind. Please check.

References are not called in a sequential way. Please change.

All references are not in compliance with the journal requirements (please check them).

References are quite complete despite not up to date. Please add more recent articles. The paper "Weldability of austenitic stainless steels by metal arc welding with different shielding gas" Procedia Structural Integrity, 2016, Vol. 2, 3508-3514 must be added for the analysis of different inert and active gas on the austenitic steel welds and embrittlement prevention.

After this major changes the manuscript can be reconsidered for reviewing.

Author Response

(The authors gave the same response as above.)

Round 2

Reviewer 1 Report

All my remarks have been included.

Author Response

Thank you very much for kind comments and help

Reviewer 2 Report

Following comments still not well responded:

1- Arrangment of manuscript is not acceptable. There is no “ Results and discussion” section! Authors are suggested to check other papers to understand how and where to add “ Results and discussion”. or you can divide it to two parts of "Results" and "Discussion" because you have already shown many results in first revision of manuscript.

2- Methodes, scope of research: “Visual test on macro samples…” Visual inspection is not a destructive test, it is non-destructive test then remove it from this part.

All items in the subsection of "The destructive tests included:" (line 212) are not destructive testing. Therefore it should be replaced with " Evaluation of microstructure and mechanical properties" because you have carried out the visual inspection, microstructure characterization and mechanical testing such as hardness and fatigue evaluations.

Author Response

(The authors gave the same response as above.)

Reviewer 3 Report

The great part of comments have been addressed.

However in the revised version (V2), at last the version received by the referee, the following comments are still open:

References are not called in a sequential way [1], [2] in the main text. Please do it.

Reference [29] is not called in the main text.

Bibliography: Reference [29] is not in compliance with the journal requirements (authors are not indicated, doi is not reported).

Bibliography: Please check accurately the format (each of them) with the journal requirements.

After this format changes the manuscript can be accepted.

Author Response

(The authors gave the same response as above.)

Round 3

Reviewer 2 Report

I still recommend to change the title in line 214 (new line number) as explained in the previous comment, however this paper is acceptable for publication in the current format.

"The destructive tests included" => " Evaluation of microstructure and mechanical properties"

Author Response

Dear Reviewer,

We have revised the article very carefully with your comments.

Thank you for your valuable comments,

 thanks to which our article has gained a lot of quality.

Reviewer 3 Report

Many comments have been addressed.

However, despite the changes, some references are still not called in the right sequence.

Ref. 5 is not called in the main text.

Ref. 11 is not called in the main text.

Lines 121-122 check font size.

Lines 169: please check the right sequence: [30, 31] before 27 and so on...

[40]: check author surnames.

After these changes the manuscript can be accepted.

Author Response

(The authors gave the same response as above.)
